# Duplex Steels Used in Building Structures and Their Resistance to Chloride Corrosion

**DOI:** 10.3390/ma14195666

**Published:** 2021-09-29

**Authors:** Mariusz Maslak, Marek Stankiewicz, Benedykt Slazak

**Affiliations:** 1Faculty of Civil Engineering, Cracow University of Technology, Warszawska 24, 31-155 Cracow, Poland; goziolko@cyfronet.pl; 2DAIKO SRL Welding Materials, Viale Felissent 84/D, 31100 Treviso, Italy; bslazak@gmail.com

**Keywords:** duplex steels, welded joints, corrosion resistance, corrosion tests, chloride environment

## Abstract

Welded structures made of duplex steels are used in building applications due to their resistance to local corrosion attack initiated by chlorides. In this paper, the material and technological factors determining the corrosion resistance are discussed in detail. Furthermore, recommendations are formulated that allow, in the opinion of the authors, to obtain a maximum corrosion resistance for welded joints. The practical aspects of corrosion resistance testing are also discussed, based on the results of qualification tests. This work is of a review character. The conclusions and practical recommendations are intended for contractors and investors of various types of structures made of the duplex steel. The recommendations concern the selection and use of duplex steels, including the issues of metallurgy, welding techniques, and corrosion protection.

## 1. Introduction

Modern two-phase ferrite-austenitic duplex stainless steels demonstrate an excellent combination of high strength and relatively high corrosion resistance [1]. Due to their fair ductility, the risk of catastrophic brittle fracture initiation is considerably limited [2]. Nowadays, the particular impulse amplifying the development of new types of duplex stainless steel is driven by increasing demand for crude oil mining from under the seabed. It translates into the need for the design of specific corrosion-resistant oil transmission installations both at offshore platforms and oil tankers. The newest, fourth generation of duplex steels has been designed recently for this purpose [3]. These materials design actions were taken to balance their microstructure and to equalize the ferrite and austenite corrosion resistance. 

The corrosion resistance testing of duplex steels has become the subject of many research programs. Various research computational methods were used and led to the exploration of possible applications [4,5,6,7,8,9,10,11,12,13,14,15,16]. However, the knowledge, despite its extensive discussion in the scientific community, has not always been effectively translated into practical recommendations for investors and designers. The authors of this article have repeatedly encountered a lack of understanding of basic issues related to the application and technology of duplex steels, which are currently also one of the most expensive products of the metallurgical industry. Therefore, we saw the need to present in a comprehensive, though simplified manner, a practical guide for the rational use of individual duplex steel grades. We hope that this paper will allow designers to optimally use the basic advantages of duplex steels. 

The authors’ observations on the progress of pitting corrosion in duplex steels are presented in this work. The tests were performed as part of the Welding Procedure Qualification Record (WPQR) certificate [17]. The testing procedure was in accordance with the guidelines of ASTM G48-11 (method A) [18] with simultaneous consideration of the acceptance criteria of the NORSOK M601: 2008 [19] and ASTM A923-03 [20] standards.

## 2. General Characteristics of Duplex Steel 

Duplex steels can be divided into five basic categories, depending on the percentage of alloying elements (Cr, Mo, Ni, Mn, Cu, and N—Figure 1). These groups are as follows:Lean Duplex Stainless Steel (LDSS);Standard Duplex Stainless Steel with 22% Cr (DSS 22% Cr);High Alloyed Standard Duplex Stainless Steel with 25% Cr (DSS 25% Cr);Super Duplex Stainless Steel (SDSS);Hyper Duplex Stainless Steel (HDSS).

A typical microstructure of properly balanced duplex stainless steel is given in Figure 2. It includes ferritice (dark) and austeniteic grains (light). The chemical composition of duplex stainless steel is limited by thermodynamic stability of austenite and ferrite and also by nitrogen solubility limit. Stability areas of duplex steel with respect to combined Cr + Mo mass fraction are given in Figure 3. Below 20% Cr + Mo, there is a risk of martensitic transformation of austenite. At above 35% Cr + Mo a δ-ferrite instability and formation of harmful secondary phases may occur. Moreover, a variation of nitrogen content in HDSS may influence the phase stability. Such difficulties are, in general, caused by high nitrogen vapor pressure. 

Balanced duplex stainless steels are theoretically located at a 50% ferrite content line in Schaeffler–DeLong diagrams (Figure 4). However, in practice in properly balanced steel microstructure, the ferrite—austenite proportions range are wider, i.e., $\frac{\mathrm{ferrite}}{\mathrm{austenite}}\approx \frac{50{\%}^{-10\%}}{50{\%}^{+10\%}}$.

The appropriate use of duplex steel demands the welding engineers perform thoughtful actions based on well-established engineering knowledge and experience. A commonly applied routine-based approach to welding admittedly leads to obtaining the expected mechanical properties of welding joints. Nevertheless, it does not guarantee the concurrent obtaining of required corrosion resistance for the joints. It is known that the corrosion resistance of joints in the chloride environment impact zone reaches only 50–80% of the parent material corrosion resistance [22].

## 3. PREN Chloride Corrosion Resistance Index

In the right of Figure 1, the values of Pitting Resistance Equivalent Number (PREN) have been given for each group of duplex steels. This index is used to evaluate the pitting corrosion resistance of the steels in a chloride environment. The PREN indicator refers to the thermodynamically stable steels, i.e., the steels after their final heat treatment [23]. In the case of duplex steels, the following formula given by Herbsleb is used to calculate the value [24]: PREN = Cr + 3.3Mo + 16N(1)

In this equation Cr, Mo, and N are weight percentages of the corresponding elements. For SDSS and HDSS, which contain W or Cu, different formulae may be used [24]. These are as follows:Okamoto formula:
PRENW = Cr + 3.3(Mo + 0.5W) + 16N(2)

Heimgartner formula:

PRENCu = Cr + 3.3Mo + 15N + 2Cu(3)

Extended formula:

PRENEXT = Cr + 3.3(Mo + 0.5W) + 2Cu + 16N(4)

In these formulae, the content of a particular alloying element is given in its mass percentage (%wt.).

The PREN values range from 26 (for LDSS steels with average pitting corrosion resistance) up to above 45 (for HDSS steels with high corrosion resistance). In both cases, the resistance is significantly higher when compared to conventional steel grades. However, the values of PREN should be treated as comparative data. The final selection of steel for a given application should be based on tests carried out in a given corrosive medium. The usefulness of the PREN indicator for estimating the analogous resistance of welding consumables is limited due to different welding techniques that can be used and hence the different levels of nitrogen introduction into the welded melt metal. The terms “high corrosion resistance” and “average corrosion resistance” should be therefore interpreted as relative measures.

## 4. Alloying Elements Influence on Duplex Steels Corrosion Resistance 

Duplex steels usually crystallize from a liquid in the form of δ-ferrite. During alloy cooling, the lattice structure stresses increase because of Fe replacement by elements with larger radii, e.g., Ni.

In the areas with a higher concentration of austenite-forming elements, the A2-type ferrite lattice structure is transformed to an A1-type lattice (with 25% larger lattice parameter) at δ-solvus temperature (Figure 5a). This phenomenon is accompanied by a decrease in tension and a simultaneous decrease in inter-granular borders energy. This is an enhancing factor for δ → δ + γ transition. Due to the presence of alloying elements, there is an increase in Cr, Si, Mo, W, P concentration in A2-type ferrite lattice and Ni, N, Cu, Mn, C in A1-type austenite lattice. The transition appears to be diffusion-limited [25]. As a consequence, the newly formed austenite assumes a lamellar (island) structure. Out of homogeneous δ-ferrite, a two-phase structure δ + γ is formed, with its components varying from one another in terms of corrosion resistance. Chrome and molybdenum (ferrite formers) strongly increase the electrochemical potential. Consequently, the corrosion resistance is concentrated in ferrite. In austenite, only an interstitial solution of nitrogen significantly can increase the electrochemical potential of steel (Table 1; Figure 5b–d).

The influence of typical alloying elements on duplex steel corrosion resistance is presented in Table 2. In general, Mn and Ni reduce the corrosion resistance. However, these elements are required due to the strengthening effect of Mn and the need for Ni to initiate a ferrite decomposition and form a two-phase structure. 

The process of δ-ferrite decomposition and formation of the δ + γ two-phase structure takes place at temperatures of 800–1200 °C. Due to its diffusive nature, the kinetics of the transformation depends on the cooling rate (Figure 6). A slow cooling causes a formation of γ-austenite in the amount close to thermodynamic equilibrium. The distribution of alloying elements between the matrix components is also close to equilibrium in this situation. The rapid cooling, on the other hand, creates a metastable structure with a lower austenite content. It increases the risk of harmful secondary phases precipitation from ferrite supersaturated with alloying additives. The formation of harmful precipitates only within ferrite is a consequence of several dozen times higher diffusion coefficients and many times lower solubility of interstitial elements (C, N) in ferrite compared to austenite [25].

To obtain a higher amount of γ-austenite in the final steel microstructure, it is desirable to cool it slowly in the temperature range of the two-phase structure. It can be ensured by a sufficiently high linear energy of welding. At a temperature below 1050 °C, the situation is reversed, and a higher cooling rate is necessary to counterbalance the high tendency to secondary phase precipitations so that the cooling line does not cross the upper Time-Temperature-Transformation (TTT) curve (Figure 7).
Figure 7Schematic diagram of TTT (Time-Temperature-Transformation) specified for the typical duplex steels (according to [31]).
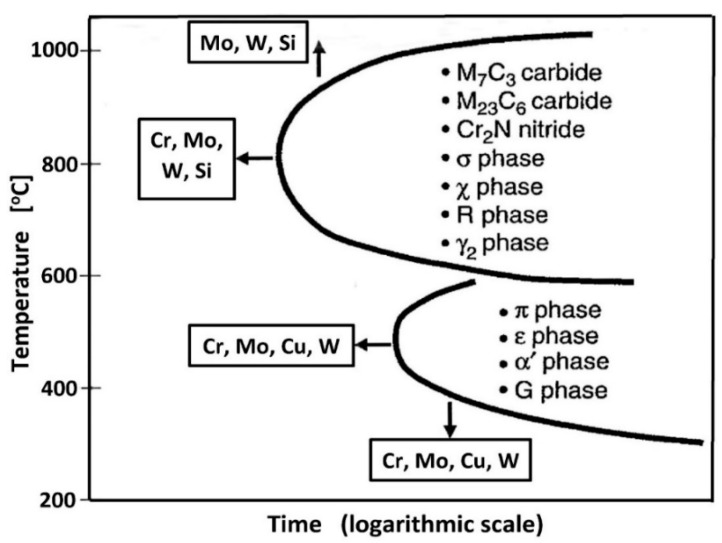

materials-14-05666-t002_Table 2Table 2Influence of various alloy additions and steel microstructure on duplex steel pitting and slotted corrosion resistance (according to [32,33]).Alloying ElementEffectCauseFormal LimitationsCNegativeA surplus of C causes secretion of chromium carbides with associated chromium-depleted zonesUp to approximately 0.03% of the contentSiPositiveStabilizes the passive top layerUp to approximately 2% due to the negative influence of Si on the stability of the structure and on the Nitrogen solubilityMnNegativeMn-rich sulfides can initiate pitting. Mn can also destabilize the passive surface layerThe content of over 2% of Mn increases the risk of precipitation of harmful intermetallic phasesSNegativeSulfides are a strong initiator of pitting corrosionHigh resistance to pitting corrosion is only possible with a content of less than 0.003% SCrPositiveStabilizes the passive top layer25% to 28% maximum depending on the Mo content. Higher Cr content increases the risk of precipitation of harmful intermetallic phasesNiNegativeIncreased Ni content lowers the PREN index of austeniteThe Ni content is limited to the amount necessary to form about 50% of austeniteMoPositiveStabilizes the passive top layer and its subsurface metal substrateAbout 4–5% Mo depending on the Cr content. Mo increases the risk of precipitation of harmful intermetallic phasesNPositiveSignificantly increases the PREN index of austeniteAbout 0.15% in LDSS steels. About 0.3% in SDSS steels and slightly over 0.4% in 25% Cr alloys, with high Mo and Mn contentWPositiveIt probably works in the same way as MoIncreases the tendency to release harmful intermetallic phasesCuUncertainMarginal effectMaximum content up to approximately 2.5%


## 5. The TTT Diagram, CPT Temperature, and the Ferrite Number Determined for Duplex Steels

In high-alloy duplex steels of older generations (SDSS and HDSS), the precipitation start time is short [34]. Exceeding the activation energy of the secondary phase precipitations, expressed by the intersection of the cooling line with the TTT curve increases the sensitivity to accumulation of heat exposure effects resulting from welding the steel. It is because the energy of the precipitation reaction is always lower than the activation energy of this process [35].

The formation of secondary phases in duplex steels, which are hard and brittle and thus harmful, takes place in two temperature ranges (Figure 7). The upper curve, corresponding to the range of 600–1050 °C, shows the precipitation of nitrides, carbides, and intermetallic phases as a result of prolonged thermal exposure of steel due to insufficiently rapid cooling. Thus, the welding with high linear energies facilitates the transformation of δ to δ + γ, which is favorable but also increases the probability of the secondary phase formation within the ferrite at insufficiently fast cooling. The consequence of this is the need to strictly adhere to the recommended linear energies of welding and to control the cooling process of the joint during welding. The lower TTT curve, corresponding to the temperature of 300–550 °C, shows the remaining secondary changes in the steel microstructure. The most important is the change of δ-ferrite to acicular secondary α′-ferrite, significantly reducing the ductility and toughness of steel. The lower limit of the occurrence of the unfavorable α′-ferrite determines the highest temperature of the long-term thermal exposure, which is about 300 °C.

The advancement of harmful changes in the microstructure, lower ductility, and corrosion resistance depends on the total heat exposure time to both critical ranges on the TTT curves (Figure 8a,b). The exposure of the lowest alloyed LDSS to temperatures between 800–1000 °C may exceed 10 h without the release of harmful phases. However, this time for the standard DSS 22% Cr steel is reduced to about 30–60 min, and for the HDSS steel is even 5–10 min [36]. The presence of 0.5% secondary phases at the boundaries of δ-ferrite causes a dramatic decrease in the breaking work (Figure 8a). The same volume of the harmful and easily released Cr_2_N phase lowers the critical pitting temperature (CPT) by up to 20 °C (Figure 8b).

Nitrogen in duplex steel increases the kinetics of austenite formation (Figure 9). With increasing nitrogen content, the high-temperature equilibrium δ + γ area expands towards lower Ni concentrations, and the temperature of austenite formation increases. It may increase even to the temperature of liquids, which means crystallization of a small part of austenite directly from the liquid metal. A significant increase in the transformation rate δ → δ + γ occurs, which, with sufficiently rapid cooling, makes it possible to avoid the effects of shifting the TTT curves on the time axis towards the origin of the coordinate system. For this reason, nitrogen-rich steels can be cooled in the temperature range of 800–1200 °C faster, without the fear of exceeding the final ferrite content above the permissible level (70%). The addition of N_2_ to the forming and shielding gases in Tungsten Inert Gas Arc Welding (TIG, 141) is of fundamental importance for obtaining a properly balanced weld microstructure.

The balanced microstructure of duplex steel is obtained in a two-stage supersaturation heat treatment. The first stage is homogenizing annealing. It is most effective at the temperature of maximum nitrogen solubility in solid solution, which, depending on the steel grade, is 1050–1150 °C (Figure 10a,b). At this temperature, harmful secondary precipitates in δ-ferrite dissolve, and the released atomic nitrogen diffuses into austenite since its solubility in the interstitial solution is many times greater (see Table 1). This applies especially to thick-walled elements made of the SDSS and HDSS steels, where the practical cooling rate in the critical temperature range is too low to avoid the precipitation of secondary phases. The second stage is rapid cooling of the steel in water, limiting the possibility of re-separation. As a result, the steel microstructure is balanced, i.e., the proportion of both components equals $\frac{\mathrm{ferrite}}{\mathrm{austenite}}\approx \frac{50{\%}^{-10\%}}{50{\%}^{+10\%}}$ and the resistance to pitting corrosion and also the KV impact strength is increased.

The volumetric fraction of δ-ferrite in the structure can be verified experimentally either by microscopic examination (image analysis) or by magnetic methods [4,5]. Values of the ferrite number (FN) are converted into the volumetric fraction of δ-ferrite (%δ) based on the following relationship [39]:%δ = 0.7FN(5)

In this equation, %δ (%) is a volumetric fraction of δ-ferrite in the steel microstructure. If it is not possible to perform the appropriate tests, the estimation of the volumetric fraction of %δ-ferrite in duplex steel can be made using the data from the metallurgical certificates in conjunction with the following relationships [36]:%δ = 4.01Cr_eq_ − 5.6Ni_eq_ + 0.016T − 20.93 (%wt.)(6)
Cr_eq_ = Cr + 1.73Si + 0.88Mo (%wt.)(7)
Ni_eq_ = Ni + 24.55C + 21.75N + 0.4Cu (%wt.)(8)

In the equations, T is the homogenizing annealing temperature, and the remaining factors are element weight fractions.

The area of the weld with the lowest resistance to pitting corrosion is the heat-affected zone (HAZ). The HAZ is particularly endangered by the time of impact near the fusion line of the temperature exceeding the δ-solvus level when the microstructure of duplex steel is a single-phase and the free growth inhibitory factor δ of the ferrite grains is missing. Lowering the temperature below the level of δ-solvus again, when the joint is cooling down, activates the δ → δ + γ reaction, which, as mentioned earlier, is a diffusion-controlled transformation. The larger the original grain size, the longer it takes to reach thermodynamic equilibrium. It is difficult to re-achieve a microstructure with a balanced proportion of both matrix phases from the developed ferrite grains. The cumulative effect of the thermal cycles of welding successive weld beads leads to an increase in the ferrite content in the HAZ at the expense of austenite. Furthermore, it leads to an increased tendency to form secondary phases from the ferrite and thus to lower both the ductility and resistance to pitting corrosion. The increased content of austenite-forming Ni in the weld and the absorption of strongly austenitic-forming N from the shielding gases intensify the kinetics of the δ → δ + γ transformation, thereby preventing the reduction in austenite content in the weld microstructure. Therefore, the selection of the parent material with the highest possible austenite content, within the limits of the correct balance of the initial duplex steel microstructure, is important for the subsequent resistance of the HAZ to pitting corrosion and its KV impact strength.

## 6. Pitting Corrosion of Stainless Steels Initiated When Exposed to Chloride Impact

Steel pitting corrosion is an electrochemical process that takes place in halide-containing electrolytes [40]. The corrosion of this type attacks the material locally and quickly perforates the metal, causing the loss of tightness. It is considered to be one of the most severe forms of corrosion. It has the highest intensity in stationary solutions due to their uneven saturation with depolarizers, i.e., oxygen or hydrogen ions, and the resulting formation of the so-called concentration cells. The intensity of pitting corrosion increases with the temperature of the electrolyte. Mixing the solution equalizes the depolarizers’ concentration at the metal surface, thereby slowing the progress of corrosion [41]. A permanent polarization of the cell, i.e., blocking the anode or cathode reaction, may stop the progress of pitting corrosion.

A simplified diagram of the steel pitting corrosion mechanism is shown in Figure 11. At structural defects, such as ferrite/austenite interfaces, non-metallic inclusions (especially sulfide), secondary phase precipitates (including carbides and nitrides), the passive layer can be easily punctured by aggressive chloride anions. A local, short-circuited corrosion micro-cell is created with a small anode area and a large cathode area. A consequence of the disproportion of the anode surface to the cathode surface ratio is a high anodic current density. Fe^2+^ ions dissolve into solution and are consequently captured by Cl^−^ anions to form iron chloride (FeCl_2_). Around the pit, a cathodic depolarization reaction takes place (oxygen-type, hydrogen-type, or both), which maintains the operation of the electrochemical cell by balancing the electric charge.

The pitting corrosion is autocatalytic, self-accelerating the growth of pits due to lowering the pH of the corrosive solution inside the growing pit [40]. In the initial stage, the pitting corrosion is of inter-crystalline character [42]. A balanced duplex steel microstructure, within the limits of $\frac{\mathrm{ferrite}}{\mathrm{austenite}}\approx \frac{50{\%}^{-10\%}}{50{\%}^{+10\%}}$ reduces the tendency to harmful precipitations of carbides and nitrides at the intergranular boundaries. This simultaneously reduces the risk of developing inter-crystalline corrosion. 

## 7. Passive Layer and Corrosion Protection Mechanisms Identified as Being Specific to Duplex Steels

The corrosion protection of Cr-Ni type stainless steels is based on a durable and tight passive layer. The passive layer should be chemically inert to an aggressive environment. The passive layer in the oxidizing environment forms itself spontaneously and reaches a thickness of 2–4 nm [43]. It has an ability to self-rebuild (re-passivation ability). The most important role is played by Cr, which forms a complex oxide (Fe, Cr)_2_O_3_ on the steel surface. The minimum content of Cr in Cr-Ni type steels to form the passive layer is 10.5%. The higher the Cr fraction in (Fe, Cr)_2_O_3_ is, the tighter and more corrosion resistant the passive layer is. Mo and N are incorporated into the passive layer in a lesser content. Molybdenum has a nobler electrode potential compared to Cr. It can form MoO_2_ and MoO(OH). Nitrogen may be present in the passive layer in the form of anions. As such, it may inhibit surface adsorption of negative Cl^−^ anions. A synergic activity of Mo and N is observed. The synergic activity increases the duplex steel pitting resistance to a greater extent than it would be expected from the cumulative content of both elements separately [44]. 

The oxidation of steel surface during welding is accompanied by migration of Cr and Mo to the surface layer. This, in turn, lowers the electrochemical potential of the metal under the passive layer and increases the risk of corrosion previously initiated. The chemical etching and passivation of the welded joints surface are an effective way to counteract this threat. Nitrogen dissolved in the steel reacts with H^+^ to form NH_4_^+^, thus partially neutralizing the acidic pH of the corrosive medium within the pits, limiting their growth [44]. It is believed that this may be the mechanism of active corrosion protection of austenite over which a thinner passive layer forms. The effectiveness of this protection is evidenced by the value of the nitrogen weighting factor in the PREN formula determined for pure austenite [23]:PREN = Cr + 3.3Mo + 30N(9)

A nitrogen deficiency in the passive layer of duplex steel limits the effectiveness of the passive layer over the austenite grains. In Figure 12a, the pits formed in the filling of the gas tungsten welded joint (TIG, 141) are shown in detail. The fillings were laid without nitrogen in the shielding gas. The joint in question was subjected to a laboratory pitting corrosion resistance test when exposed to FeCl_3_ solution. A fragment of extensive corrosion pitting was selected for analysis. The alloying elements’ distribution in the part of the pitting surface indicated by a white arrow revealed a relationship between the distribution of Cr in the surface layer and Cl absorbed from the corrosive solution. In places with a high Cr concentration (the surface layer above the ferrite grains), the content of absorbed Cl on the surface was low. On the other hand, in places with a low concentration of Cr (the surface layer above the austenite grains), the amount of absorbed Cl was high. It is important to note that the high concentration of absorbed Cl occurred at places with the highest corrosion intensity. In the case considered here, the SDSS weld metal had low N content and, therefore, an insufficiently balanced microstructure. The preferred corrosion attack was on the subsurface austenite grains and then, due to the presence of harmful secondary phases, on the adjacent grain boundaries. Subsequently, the grains of both basic phases of the metallic matrix were etched.

## 8. Recommendations on the Chemical Compositions of Duplex Steel Grades with Improved Corrosion Resistance 

Because of the different physico-chemical properties of the passive layer over the ferrite and austenite areas, the balanced microstructure reduces the risk of pitting corrosion. Nowadays, the metallurgical industry offers duplex steel products with a microstructure balance within the limits of $\frac{\mathrm{ferrite}}{\mathrm{austenite}}\approx \frac{50{\%}^{-10\%}}{50{\%}^{+10\%}}$. However, due to the risk of excessive ferritization of the heat-affected zone (HAZ) on the welded joints, the parent material should be used with the δ-ferrite content not exceeding 55% [45,46,47,48,49].

In each type of steel, the presence of carbon and sulfur precipitates with high surface energy facilitates the initiation of corrosion. For this reason, the selection of materials with the lowest C and S contents possible is recommended. The presence of atomic oxygen, dissolved in the solid steel solution, supports electrochemical corrosion due to its role in the cathodic reaction. It is therefore advisable to deeply deoxidize the liquid metal [50].

It is recommended to use the fourth-generation duplex steel. Welding of the high-alloy duplex steels (both SDSS and HDSS) of the previous generations was associated with technological difficulties to ensure an appropriate cooling time (Δt_12/8_) necessary for the proper balance of the weld microstructure. Increased kinetics of austenite formation and reduced sensitivity to selective corrosion of the δ-ferrite in the fourth generation steel softens the welding technological regime necessary to obtain the required resistance to pitting corrosion of such welded joints [22].

Duplex steels should not be welded without additional material due to the risk of excessive ferritization of the weld, resulting from melting of the parent material. The weld metals should have a Ni content higher by 2–4% compared to the parent material in order to increase the kinetics of austenite formation δ → δ + γ as well as to compensate for the lack of subsequent balancing of the welded joints microstructure by heat treatment. For this reason, it is recommended to weld the lower grades of duplex steels with the weld metal of the composition corresponding to the higher grade (e.g., DSS steels should be welded with the weld metal of the chemical composition of SDSS steels). This applies in particular to the weld root beads exposed to direct contact with the corrosive medium. Examples of auxiliary materials dedicated to welding various grades of duplex steels are given in [51]. 

Nitrogen present in most duplex steel grades enhances the kinetics of austenite formation. The relationship between the nitrogen content and δ-ferrite content in the weld is linear (Figure 13).

The high vapor pressure of nitrogen at the welding temperature causes its migration from the weld pool to the surrounding environment and, consequently, reduces the share of austenite in the weld structure, thereby increasing the risk of losing corrosion resistance and lowering the impact strength.

The nitrogen-poor weld contains up to 80% δ-ferrite. Since the electric arc does not transfer the electrically neutral nitrogen atoms, N is not present in arc welding consumables. Therefore, the only option to increase the N content in the weld metal during the welding is the addition of N_2_ to the shielding and forming gases. The necessary amount of N_2_ depends on its solubility limit in duplex steel and increases with increasing Ni concentration in the steel. Nitrogen content in shielding gases for GTAW welding (TIG, 141) should be in the range of 1–1.2% for DSS 22% steel and 2–2.5% for DSS 25%, both SDSS and HDSS steels.

The weld root is usually the area with the greatest risk of corrosion. Therefore, it is important to use N_2_-rich mixture as forming gas. However, the use of shielding gases with an excessively high N_2_ concentration may result in exceeding the N solubility limit in solid solution and the appearance of weld porosity, especially in thick-walled joints. Since the corrosion resistance of stainless steel is primarily determined by the properties of the surface layer, the beads of the multi-run welds can be welded in pure argon to avoid porosity. In such cases, the pitting corrosion resistance tests should not include the weld filling. 

The addition of 20–40% of helium to shielding gas increases the thermal energy supplied to the weld, and this allows increasing the welding efficiency with the GTAW method (TIG, 141). Furthermore, the full control of the O_2_ content in welding gases prevents its absorption in the weld pool, as well as a harmful O increase in the solid solution. In addition, it allows a reduction in the thickness of the oxide layer above the welded joint and thus the depth of the depletion of the steel surface layer in Cr and Mo. Therefore, it is recommended to use welding gases with O_2_ content below 200 ppm for duplex steels and to flush the pipes from the weld root side with forming gas in order to reduce the O_2_ content as much as possible. It is suggested to limit the O_2_ concentration to 25 ppm.

## 9. Recommended Welding Technologies

Expensive duplex steels are primarily used because of their high corrosion resistance in chloride environments. The correct welding technology selected for these steels should therefore ensure sufficient corrosion resistance of at least those areas of welded joints that remain in contact with the aggressive medium. In the case of single-sided welding, e.g., pipelines, small vessels, and containers, it is usually the weld root with the adjacent heat-affected zone that is most susceptible to corrosion. Whenever possible double-sided welds should be designed since balancing the microstructure and achieving the required level of resistance to pitting corrosion of the weld face is much easier than with the weld root. In single-sided joints with an accessible weld root, the backing weld of the root may be used to improve the low corrosion resistance. To each bead of the weld, it is necessary to introduce the appropriate amount of thermal energy, limit the access of oxygen, provide the necessary time for decomposition of δ-ferrite, and for the formation of an optimal amount of γ-austenite. This is done by slow cooling between the 1200 °C and 1050 °C, and the release of harmful secondary phases is prevented by quick cooling between the 1050 °C and 300 °C. The temperature range of 1050 °C to 850 °C requires a particularly intensive cooling of the steel. In our opinion, the optimal heat amount, introduced into the weld to ensure the expected cooling rate above and below 1050 °C, seems to be the fundamental issue in welding duplex steels with classic arc methods. Moreover, the welding of subsequent weld beads may not cause adverse effects in the microstructure of the preceding weld beads, especially in the areas of the weld root, which are in direct contact with the corrosive environment. Therefore, to ease the control and improve the uniformity of the heat transfer, it is recommended to use mechanized welding instead of manual welding.

Limiting the access of oxygen to the root of the weld requires the use of low-oxygen welding methods. In Figure 14a, the relationship between the breaking energy KV of the weld metal and its oxygenation is shown in detail. The lowest degree of oxygenation in the weld metal is achieved by using the GTAW method (TIG, 141) and PAW (Plasma Arc Welding) method (151), which is a GTAW method extension.

The ease of GTAW (TIG, 141) made this method the primary choice. For the pipe connections, a modified GMAW-STT method (Surface Tension Transfer, MIG-STT, 131-STT) may be used. It provides a three to four times higher welding efficiency in rela-tion to the conventional GTAW and gives a comparable resistance to pitting corrosion. Additionally, it guarantees a satisfactory ductility of the material down to −40 °C [53]. It should be noted that the use of high-oxygen, slag arc welding methods for this type of welds, such as SMAW (111), SAW (121), or FCAW (114), reduces the corrosion resistance of the joints and also reduces the breaking energy of the weld metal. At the same time, it increases the lower operating temperature threshold for welded joints use (Figure 14b). This is a consequence of the high content of atomic oxygen dissolved in the solid solution and the presence of oxide inclusions at the grain boundaries.

As most standards do not require pitting tests of the whole weld cross-section, any flux-type welding (high-oxygen) methods can be applied for inner layers of thicker multi-run joints. However, particular caution should be taken while welding SDSS and HDSS steels due to their high yield strength and a stronger tendency to brittle fracture. In such a situation, an application of the low-oxygen welding methods (both GTAW and PAW) at the whole cross-section of the weld and precise balancing both of the weld microstructure and the whole heat-affected zone (HAZ) is required for obtaining reasonable impact resistance. 

The regulation of γ-austenite formation kinetics in duplex steel welds is dependent on the proper shaping of the weld groove. The shapes of the grooves should be in general analogous to those formed in acid-resistant austenitic steels. Nevertheless, minor discrepancies are possible in the current situation. Examples of typical grooves recommended for welding duplex steels are presented in [38]. For single-sided welding, the grooves should be shaped to obtain a wider root gap, lower root face, and wider groove angle (bevel) [46]. The wider root gap and lower root face limit the weld metal and the parent material mixing rate, which reduces the Ni content in the weld metal of the weld root. The root bead should be massive enough to counteract the nitrogen deficiency at this welding step by extending the cooling time in the austenite formation temperature range. The weld root should be welded using high linear energy, within limits recommended by the weld metal manufacturer. Under-heating of the weld root bead accelerates the cooling in the austenite formation temperature range. Nevertheless, excessive overheating lengthens the cooling time and stimulates the precipitation of harmful secondary phases. The consequence is a reduction in pitting corrosion resistance and impact strength. The following filling beads are often called “cold” runs. They should be welded with the linear energy reduced by up to 75% and should not be massive so as not to cause changes in the microstructure of the root run and in the HAZ, reaching directly under the passive layer. The following filling runs are to be welded with recommended increased heat input energy, which in the face layer reaches up to 150% of the heat input used for the weld root [54]. The thermal effects of the successive layers of the weld, lying above the “cold” run, must in no way affect the microstructure and properties of the weld root run.

The most problematic for maintaining the proper microstructure and sufficient pitting resistance seems to be the single- and two-runs of the thin-walled welds. Such problems appear in the seal welds of shell-and-tube heat exchangers [55]. Delicate girth welds in-between massive perforated bottom and thin-walled pipe are often welded with an intense mixing rate of weld and parent material of the pipe, and additionally, the cooling rate is higher due to massive perforated bottom. The content of δ-ferrite usually exceeds the limiting value of 70% even when using recommended welding material. In such cases, the use of austenitic weld metal with a high Mo content allows obtaining ferrite amount slightly smaller than 70%, which means a limited resistance to pitting corrosion. Furthermore, due to the cumulative effects of heat exposure of duplex steel and precipitation of secondary phases, it is not advisable to cut the materials thermally. In order to avoid a heat accumulation during welding, it is not recommended to preheat the steel, apart from drying the surface at a temperature not exceeding 100 °C. For the same reason, the inter-pass weld temperature should be strongly limited. 

The recommendations given above allow obtaining welded joints resistant to pitting corrosion for standard duplex DSS 22% Cr steels. However, with increasing Cr content, the necessary cooling rate must be controlled. An insufficient austenite content in the weld can occur if the cooling rate is too high. A release of harmful secondary phases can be observed if the cooling rate is too low. The welding of high Cr duplex steels can be facilitated by using a combined welding method, for example, GTAW with assisted cooling of the weld by a micro-jet injector and argon as a refrigerant [56,57]. The rapidly expanding gas quickly removes heat from the weld, allowing for 2–3 times higher intensification of the cooling process. The introduction of micro-jet cooling allows increasing the welding heat. It ensures an increase in the share of austenite in the microstructure and thus increases the pitting corrosion resistance of the joint. Moreover, the controlled and appropriately rapid cooling below 1000 °C avoids the formation of harmful secondary phases. 

The implementation of the high-temperature heat treatment after the welding, carried out to properly balance the microstructure of duplex steel joints, is possible only for small objects that can be fully homogenized annealed at 1050–1150 °C and then supersaturated in water. Local heat treatment with the use of heating mats cannot be used due to degradation of the steel microstructure at the edges and due to the impossibility of rapid cooling. The large objects can be stress-relieved by tempering at temperatures below 300 °C for about 10 h so as not to initiate the microstructural changes within the lower TTT curve, as shown previously in Figure 7.

The corrosion resistance of welded joints made of duplex steel can be improved by chemical etching. The etching removes the oxide layer formed above the weld and heat-affected zone as a result of welding and re-establishes a more compact passive layer. The original oxide layer on the joint and in HAZ can be thick up to 100 nm [54]. The layer is enriched with Fe_2_O_3_ and thus has a low resistance to pitting. The large depletion layer of Cr and Mo further facilitates the development of pitting corrosion. The etched surfaces are usually treated with highly oxidizing nitric acid, hydrofluoric acid, or peroxide [58]. The chemically formed passive layer is tighter, and the concentration of Cr_2_O_3_, MoO_2_, and MoO(OH) in the layer is higher. 

## 10. Additional Requirements for Welding Quality Control

Ensuring the corrosion resistance of duplex steel welded joints requires extending the conventional routine quality control activities with a few additional measures. The first is the need to control the O_2_ content in shielding and purge gases. The concentration should be kept below 200 ppm O_2_. This can be achieved by sufficient gas purging flow of the inside area. The second requirement is the need for continuous and accurate monitoring of the weld inter-pass temperature during the welding. In duplex steels welds, this temperature is usually significantly lower than in other steels [59]. An overheating may cause the precipitation of harmful secondary phases. Moreover, it is necessary to constantly monitor the ferrite content with a ferritometer. This is especially important in the case of single- and double-run of thin-walled welds, as they have an increased tendency to excessive ferritization.

The changes in the microstructure of duplex steel joints are usually accompanied by a decrease in the breaking energy KV measured in the impact test. Low values of the toughness KV of the considered weld, or the whole HAZ, tested according to the standard ASTM A923 (Method B) [20], may indicate a possible lack of sufficient corrosion resistance. In this case, it is recommended to perform specialized pitting corrosion resistance tests in the chloride environment. The final control of the passivity of such welded joints, both in installations and in structures made of stainless steel, may be performed after the final etching and passivation using portable testers such as, for example, the Oxyliser 3 probe [60].

## 11. Corrosion Resistance Tests of Welded Joints Made of Duplex Steel, Carried out for the Chloride Environment

The American standard ASTM G48 [18] is the leading standard for corrosion resistance testing of duplex alloys and their welded joints. It contains several fundamental test procedures for assessing the resistance of stainless steels and related alloys in a ferric chloride solution. However, in the ASTM G48 standard, the criteria for evaluating the test results obtained after the experiments are not explicitly defined. Therefore, the results should be interpreted in conjunction with other guidelines taken, for example, from the Norwegian standard NORSOK M601 [19] or American standard ASTM A923 [20]. The corrosive environment in these tests is an oxygenated aqueous 6% FeCl_3_ solution. This salt partially hydrolyzes in water. The temperature of the solution increases the degree of hydrolysis, which results in a more acidic solution. The FeCl_3_ salt does not introduce foreign metal cations into the corrosive environment. The solution is not oxidizing. As such, it does not passivate the metal surfaces and has a high penetration capacity for the surface micro-damages [61].

The results of FeCl_3_ tests are affected by temperature and autocatalytic course of pitting corrosion. According to the ASTM G48-method A standard procedures, the recommended test temperature for duplex DSS 22% Cr steels is 22 ± 2 °C and for SDSS steels is 35 ± 2 °C. The use of thermostatic water baths with temperature stabilization at ±0.2 °C is recommended. If there is no consensus on the temperature conditions of tests, a deviation from recommendations of the ASTM G48-method A standard is permissible, provided that all the other elements of the standardized test procedure are followed. This deviation is allowed since in less corrosive environments, such as, for example, NaCl solution, the test temperature may be correspondingly higher. Due to the autocatalytic nature of the pitting corrosion, the extension of the test duration is accompanied by an increase in the average daily mass loss. Then, there is a gradual blurring of differences in corrosion losses between materials of different resistance, and the probability of obtaining an unreliable test result increases. For these reasons, the test time originally proposed in ASTM G48-method A (72 h) has been reduced to 24 h in the NORSOK M601 and ASTM A923 standards. The standards recommend using flat samples with dimensions of 25 mm × 50 mm or sections of tubular surfaces which are equivalent to these sizes. Any unevenness caused by machining should be smoothed and sharp edges rounded. Moreover, efforts should be made to minimize the side surfaces of the samples. In the case of thick samples, taken, for example, from multi-pass joints, cutting a thin sample from the weld root or weld face layer can be a good option as it is responsible for the corrosion resistance of the entire joint. Ideally, the exposed surface should be representative of the corrosion risks within the joint. It must therefore encompass the joint itself, HAZ, and base material. It is also advisable to mirror the surface roughness of the welded joint. The root and face weld surfaces should not be mechanically polished [62]. In the comparative tests of pitting corrosion resistance of the basic materials, a maximum standardization of the shape, dimensions, and surface conditions of samples must be guaranteed. It is also necessary to round the sample edges.

The NORSOK M601 standard supplements the requirements with preliminary etching in HNO_3_ and HF solutions. By etching, a thick and leaky passive layer above the weld and heat-affected zone is removed. A re-passivation under free oxidation in the air requires at least 24 h to obtain a sufficiently thick and tight passive layer. The NORSOK M601 standard suggests maintaining the time interval between the sample preparation and the test itself. Direct corrosion testing immediately after etching may result in uniform corrosion without visible pitting, exceeding the limit of the allowable weight loss. In such a case, it is necessary to repeat the corrosion test by doing the preparations again and keeping a 24 h interval between the HNO_3_ + HF pre-etch operation and the main FeCl_3_ test.

It is permissible and beneficial to replace the manual sample washing with an ultrasonic bath. This ensures more effective removal of corrosion products from the sample surface and positively affects the reliability of mass measurements. The test should be performed in a stationary medium with free air access to the FeCl_3_ solution. Cutting off the access of oxygen and solution stirring causes the polarization of corrosion cells and the electrochemical processes between the sample and the solution may cease.

The results of the pitting resistance test according to ASTM G48-method A are assessed based on the weight loss measurements and visual inspection. Due to the small weight loss of the samples during the test, analytical balances with a measurement accuracy of 1 mg should be used. The permissible maximum weight loss, according to NORSOK M601, is 4 g/(m^2^ per day). It corresponds to a corrosion rate of ~0.2 mm/year. In the tests associated with ASTM A901; however, a weight loss of 1 g/(m^2^ per day) is allowed, which corresponds to a uniform corrosion rate of ~0.05 mm/year. In practice, these thresholds are achieved at the time of few tiny pit appearances on the sample surface, noticeable at 20× magnification. The occurrence of a single pitting on the surface of a sample, noticeable 20× magnification, qualifies the test result as negative.

In Figure 15 and Figure 16, typical visual assessment results are shown [21]. The tests were carried out according to procedures recommended for use in the ASTM G48-method A standard. The photos presented in Figure 15 show the classic DSS 22% Cr steel, grade 2205 (1.4462, F51), hand-welded with the GTAW method (TIG, 141), shielded with Ar + N_2_ mixture, with 2209 wire. A well-balanced parent material with δ-ferrite content of ~51% was used for welding. As there is a relatively long initiation time of secondary phase precipitation, it is possible to extend the thermal exposure time in the range of austenite forming temperature by introducing more heat to the weld. As a result, ~42% δ-ferrite content has been obtained in the root of the weld, and ~55% in its face. The thermal welding cycle increased the ferrite content in the HAZ to ~62%. Due to the relatively low temperature, the amount was still within safe limits. The cooling rate below 1050 °C was sufficiently high, and no harmful secondary phases were released at the ferrite grain boundaries. The corrosion resistance test performed according to the ASTM G48-method A procedure, in conditions typical for the DSS duplex steel, showed a weight loss of less than 1 g/(m^2^ per day).

The photograph of a representative sample surface, taken with a 20× magnification (Figure 15), does not reveal any signs of pitting corrosion. Similarly, no corrosion changes were found on the weld root side. It shows that in the case of DSS 22% steel, the compliance with the material and technological recommendations allows an average experienced welder to obtain a joint with the required level of pitting corrosion resistance.

The opposite situation is shown in Figure 16. The SDSS duplex steel, grade 2507 (1.4410, F53), was subjected to manual GTAW welding (TIG, 141), shielded with Ar + N_2_ mixture, with 2509 wire. Nevertheless, either an insufficiently balanced or heterogeneous base material was used for welding with increased δ-ferrite content, amounting to ~63% in the base material and to ~55% in the HAZ. It can be presumed that the heat introduced into the joint was insufficient to form an optimal amount of austenite. This is indicated by the high content of δ-ferrite, both in the root and in the face of the weld, amounting to ~70%, and by the small size of austenite dendrites. The representative photo of a sample surface, taken from the weld face side with 20× magnification, reveals intense pitting corrosion in the weld face at the fusion line (Figure 16). It may be a result of a too-short cooling time, low Ni content in the welding wire, use of shielding gas with low N_2_ content, or a manual error in welding, causing excessive local mixing of the parent material and weld metal. The corrosion resistance test, carried out according to the ASTM G48-method A procedure, showed a weight loss of over 10 g/(m^2^ per day). This example shows the sensitivity of SDSS steels to welding technology errors, resulting in a lack of resistance to pitting.

## 12. Concluding Remarks 

This work was meant to be a practical guide for designers, contractors, and investors of various types of structures made of duplex steel. It provides a set of recommendations. The suggestions presented here result from many years of professional experience of the authors in the field. Our observations show that many mistakes are still made, mainly due to the incorrect selection of steel or an inappropriately selected welding technology.

The basic criterion for assessing the suitability of a given steel grade was its resistance to pitting corrosion. Sufficient corrosion resistance can be achieved by:

careful selection of the parent material with the lowest possible content of ferrite-forming elements, the lowest possible content of C and S in the melt and properly balanced steel microstructure within limits $\frac{\mathrm{ferrite}}{\mathrm{austenite}}\approx \frac{50{\%}^{-10\%}}{50{\%}^{+10\%}}$;use of the low-oxygen welding methods, protection of the weld pool against oxygen absorption from the atmosphere;use of a dedicated additional material with an increased Ni content (2%–4%), and ensuring the enrichment of the weld metal with austenitic nitrogen by using forming and protective gases with an appropriate N_2_ content;regulating the kinetics of phase transformations during welding by appropriately shaping the welding groove, controlling the amount of heat introduced into the weld and massiveness of subsequent beads;avoiding harmful thermal exposure at temperatures exceeding 300 °C, including adherence to the recommended inter-pass temperature during welding and its continuous monitoring;monitoring the level of δ-ferrite content in the joints;chemical etching and passivation of both the weld area and adjacent HAZ.

The application of a joint cooling method, with the use of a micro-jet injector in mechanized welding, may support the regulation of phase changes kinetics in the high alloyed DSS 25% Cr, SDSS, and HDSS steels. It enables welding with high linear energies of the arc, which positively influences the δ-ferrite decomposition and formation of austenite in the weld. It provides a fast cooling of the joint within the safe temperature range and outside the harmful phase transformations of both the steel and duplex weld metal.

The pitting corrosion resistance tests in chlorides should be carried out in environmental conditions that do not exceed the limiting resistance of the tested materials. The tests should be conducted on those areas of the welded joint that are essential for corrosion protection in the specific installation of the welded structure.

## Figures and Tables

**Figure 1 materials-14-05666-f001:**
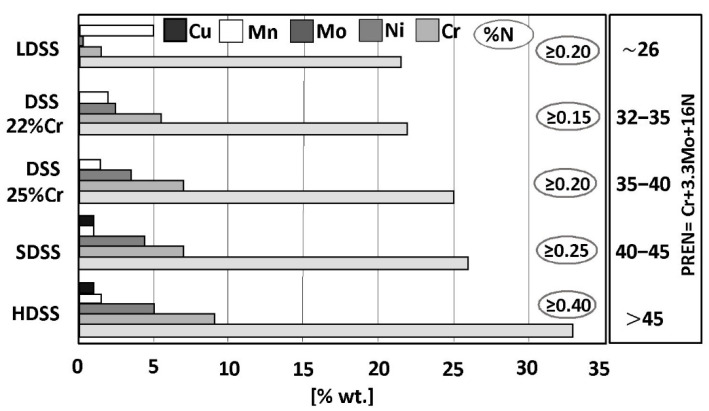
Chemical compositions and Pitting Resistance Equivalent Numbers (PREN) for various categories of duplex steels—according to [21].

**Figure 2 materials-14-05666-f002:**
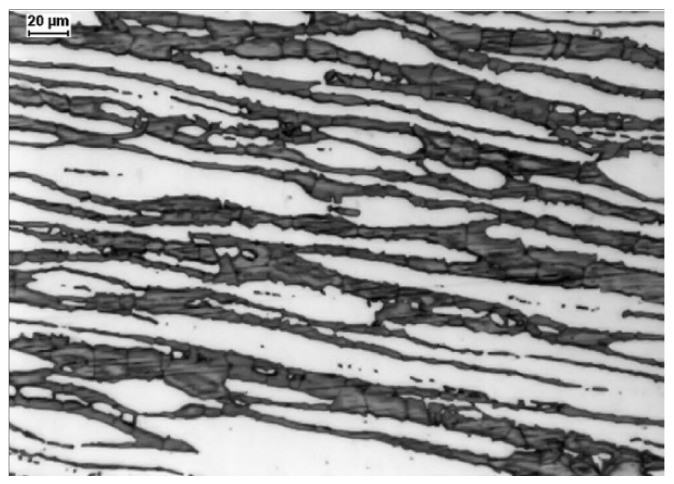
A properly balanced duplex steel microstructure containing approximately 50% of each δ-ferrite (dark areas) and γ-austenite (light areas)—according to [21].

**Figure 3 materials-14-05666-f003:**
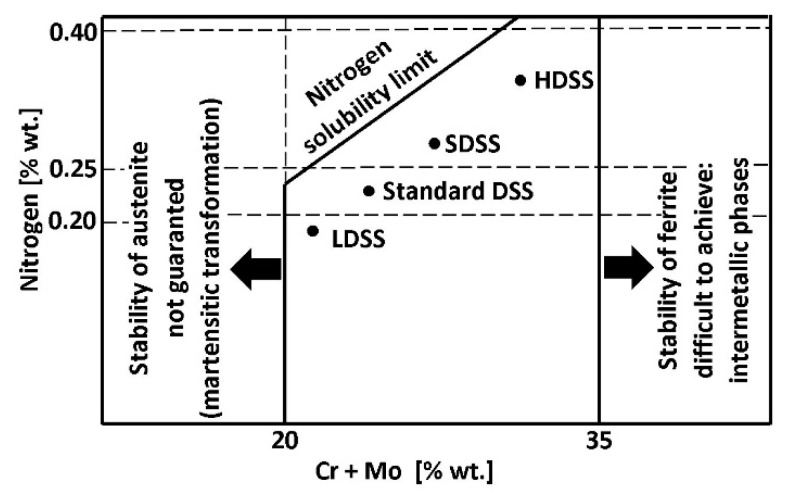
Stability areas of conventional duplex steels—according to [21].

**Figure 4 materials-14-05666-f004:**
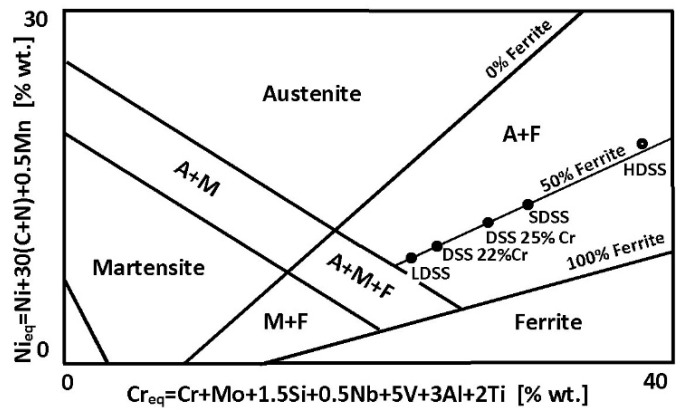
Location of individual categories of duplex steels in the Schaeffler–DeLong diagram—according to [21].

**Figure 5 materials-14-05666-f005:**
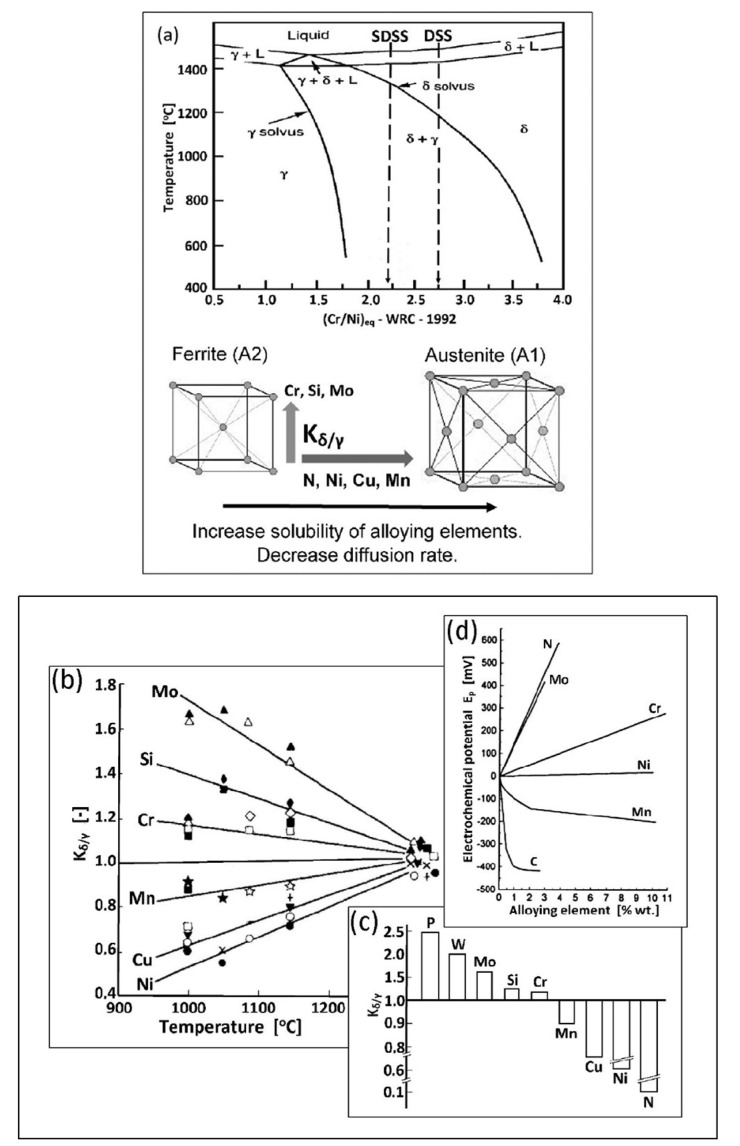
Metallurgical transformations in duplex steels: (**a**) phase equilibrium diagram and austenitic transformation δ → δ + γ (according to [27]), (**b**) distribution coefficient of alloy additions K_δ/__γ_ as a function of temperature (according to [28]), (**c**) typical values of the partition coefficient of alloying elements K_δ/γ_ for supersaturated and water-cooled steels (according to [28]), (**d**) influence of alloying elements on the size of electrochemical austenite potential in stainless steels 18/8 (according to [29]).

**Figure 6 materials-14-05666-f006:**
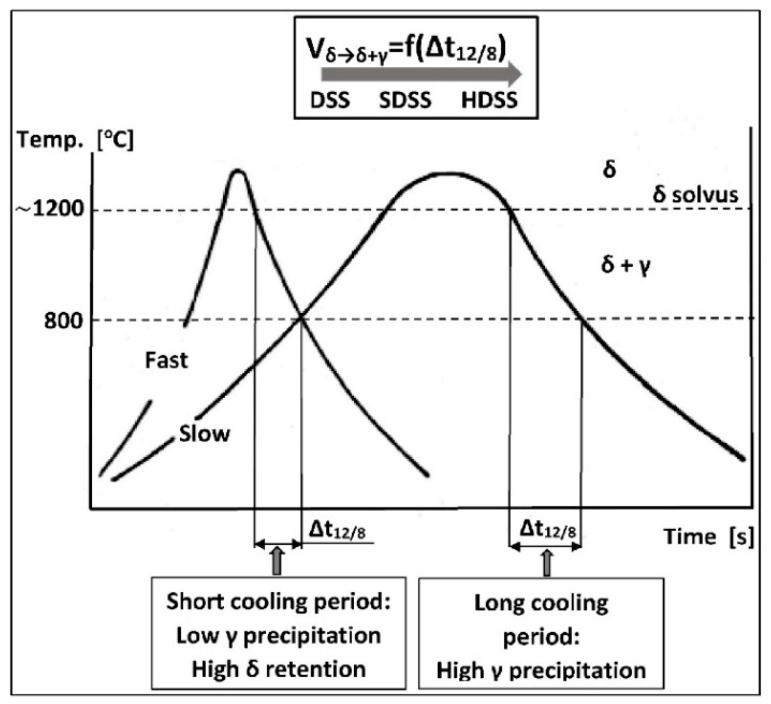
Influence of cooling rate on the content of δ-ferrite in duplex steels (according to [30]). Left-hand frame: fast cooling, high δ-ferrite content; right-hand frame: slow cooling, high content of γ-austenite precipitates.

**Figure 8 materials-14-05666-f008:**
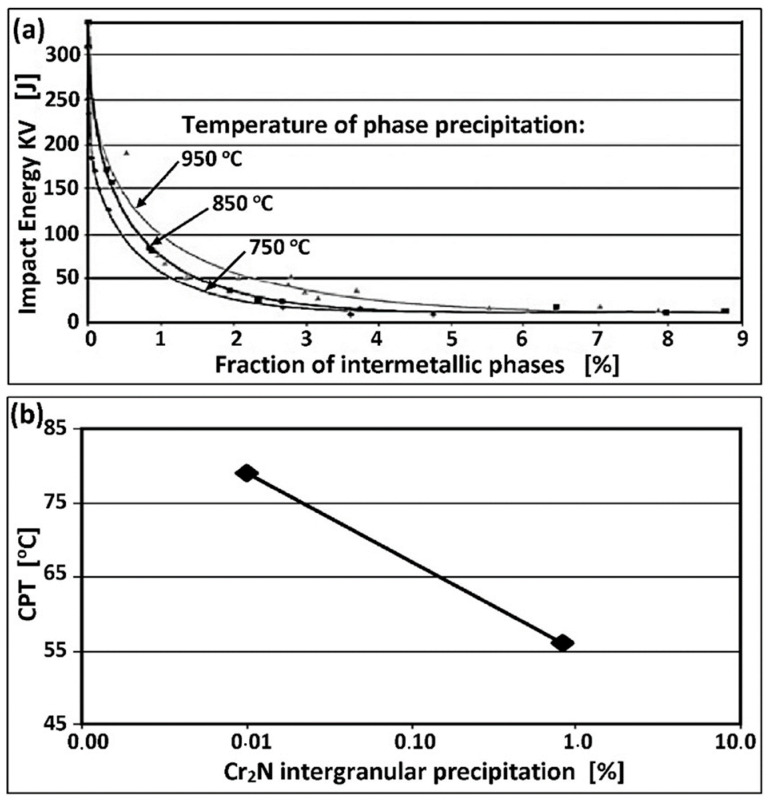
Changes in the microstructure lowering ductility and corrosion resistance of duplex steels, including (**a**) breaking energy KV as a function of intermetallic phase content in duplex steels (according to [37]), (**b**) CPT (Critical Pitting Temperature) index as a function of the content of intercrystalline Cr_2_N precipitates (according to [38]).

**Figure 9 materials-14-05666-f009:**
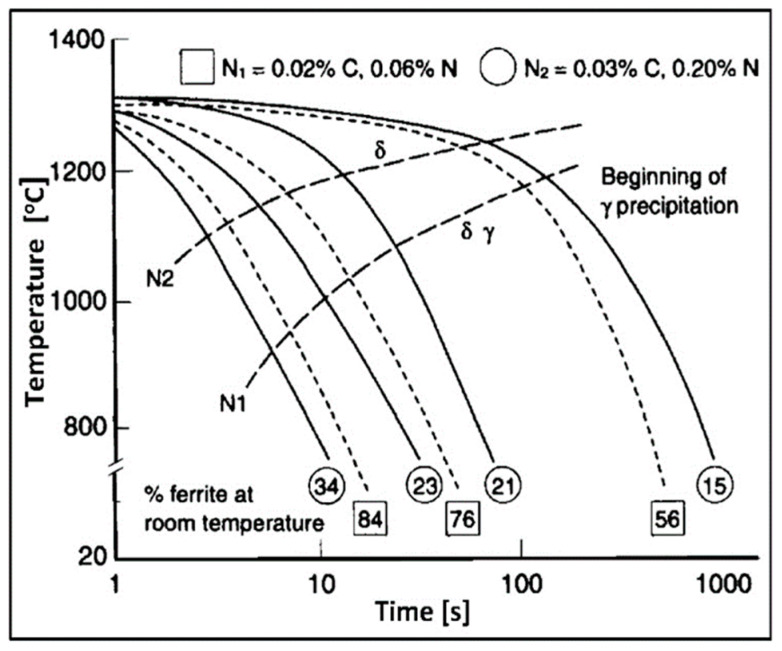
TTT diagram for DSS 25% Cr duplex steels with different alloyed nitrogen content (according to [28]).

**Figure 10 materials-14-05666-f010:**
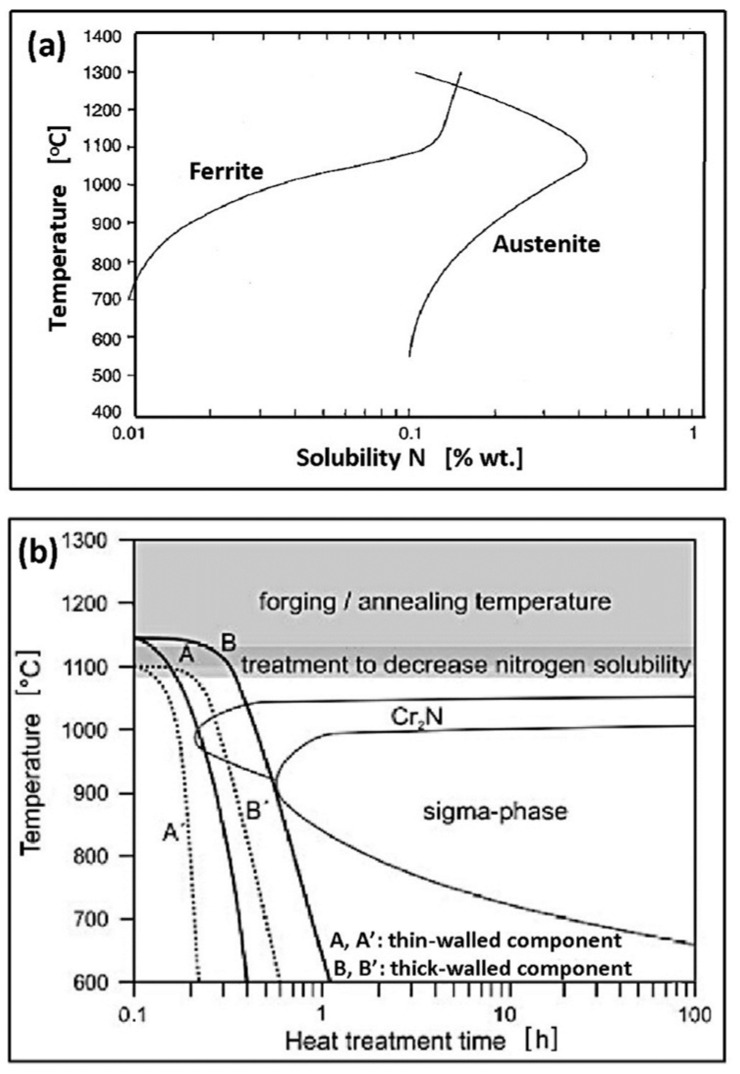
Heat treatment of duplex steel, including (**a**) solubility of Nitrogen in ferrite and austenite (according to [38]), (**b**) heat treatment of the thin-walled and thick-walled duplex steel elements (according to [23]).

**Figure 11 materials-14-05666-f011:**
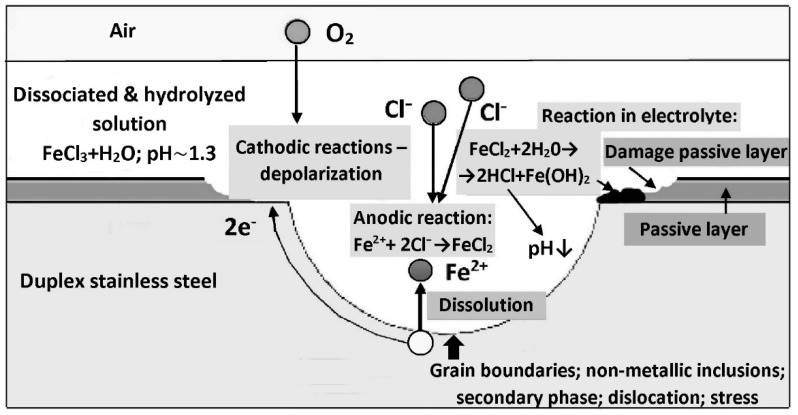
Pitting corrosion mechanism of duplex steel when exposed to chloride attack (according to [21]).

**Figure 12 materials-14-05666-f012:**
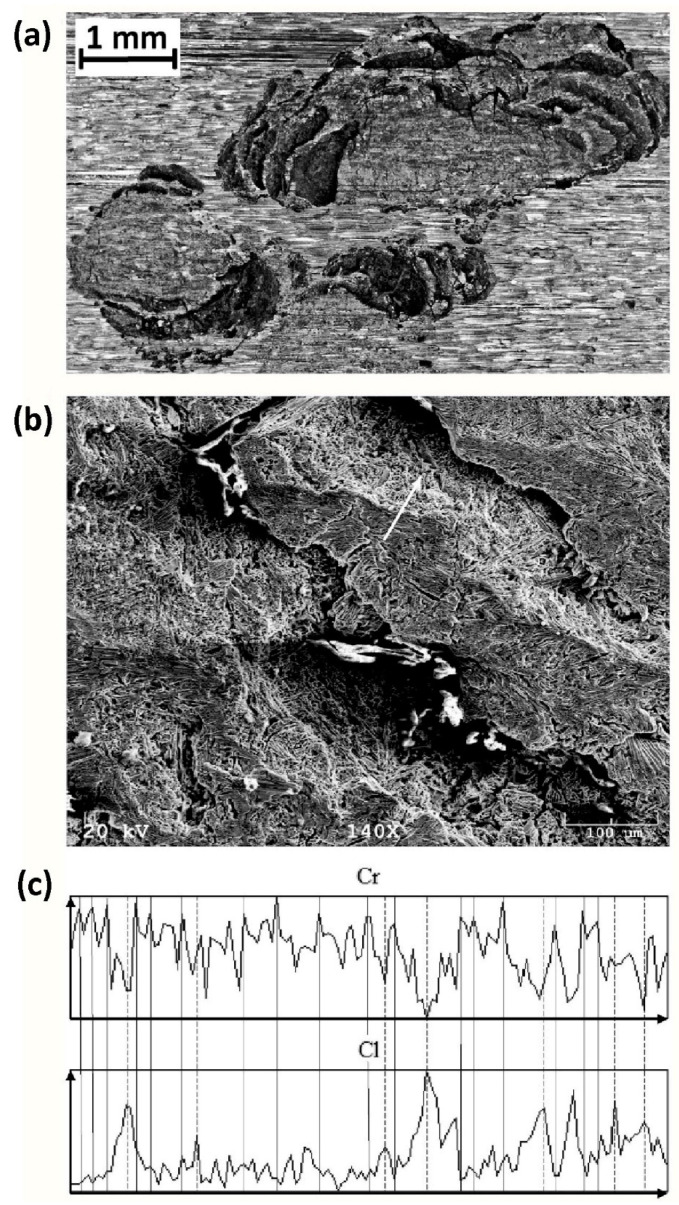
Chloride pits identified in the SDSS weld metal (1.4501, F55). Joint filling obtained after automatic TIG welding in Ar shielding gas without N_2_. In particular: (**a**) magnified optical photography, (**b**) a photo taken with a scanning microscope, (**c**) Cl and Cr contents measured along the white arrow marked in Figure 12b. The following locations are marked by vertical lines in Figure 12c: solid lines—local maxima of the Cr content in the surface layer and the corresponding local minima of the absorbed Cl content, dashed lines—local minima of the Cr content in the surface layer, and the corresponding local maxima of the absorbed Cl content.

**Figure 13 materials-14-05666-f013:**
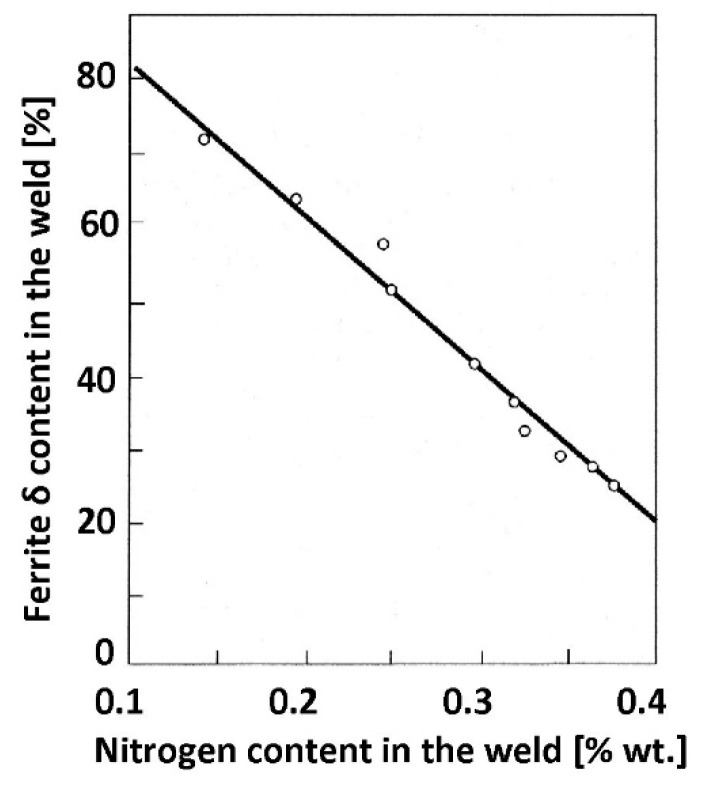
Influence of nitrogen content in a weld made with the TIG method on the volume fraction of δ-ferrite in the steel microstructure (according to [52]).

**Figure 14 materials-14-05666-f014:**
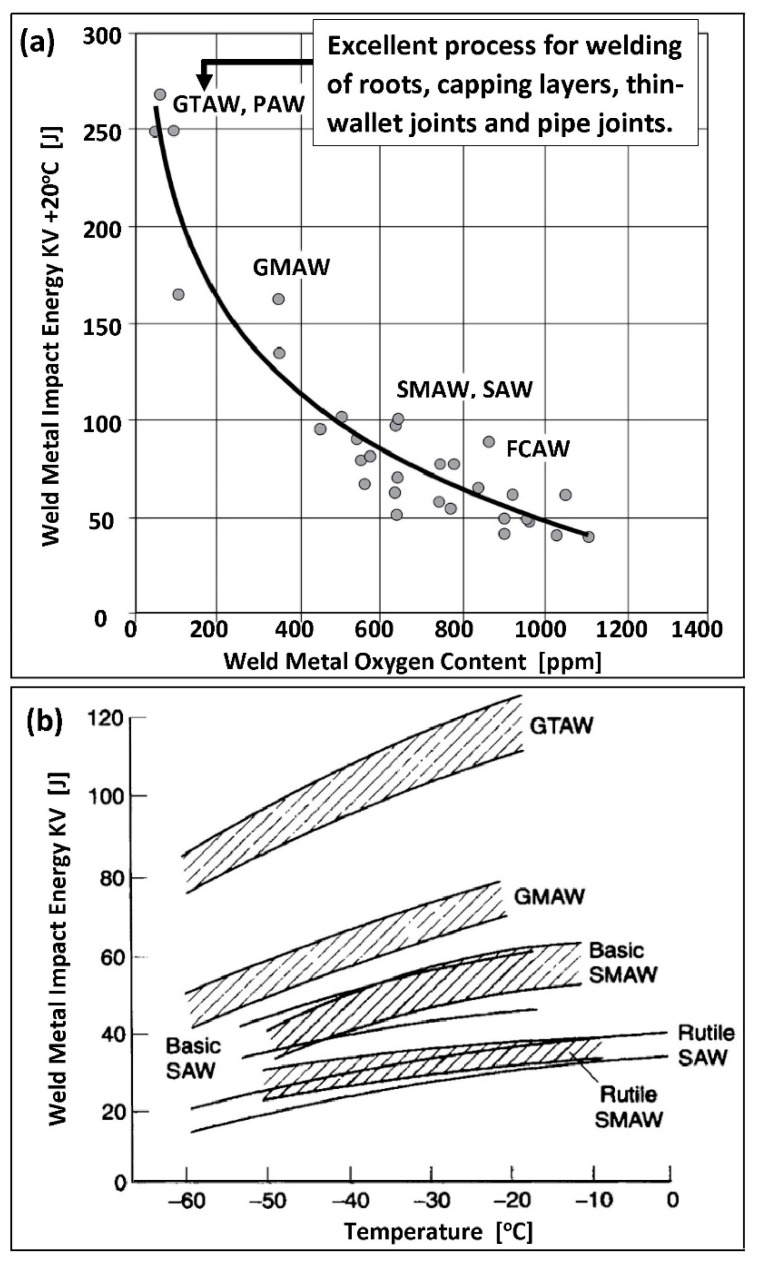
Breaking energy of the weld metal: (**a**) breaking energy KV determined at 20 °C as a function of oxygen content in the duplex weld metal, (**b**) breaking energy KV determined at 0–(−60) °C, specified for various welding methods (according to [28]). The meaning of individual abbreviations is explained in the Abbreviations of this paper.

**Figure 15 materials-14-05666-f015:**
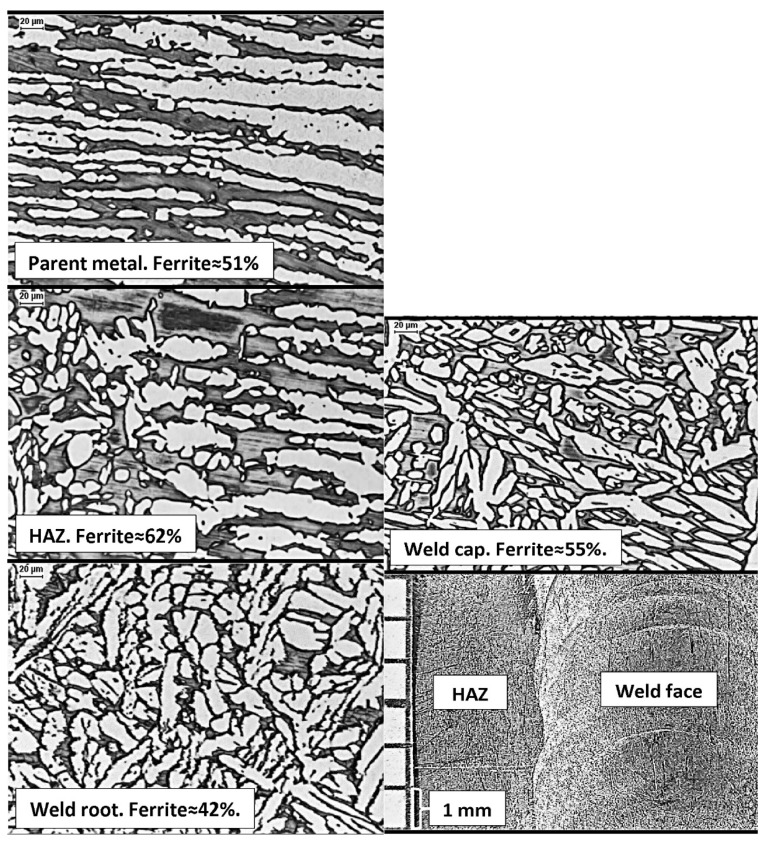
Authors’ analysis of welded connection made of duplex DSS steel with high resistance to pitting corrosion identified for the environment of chlorides.

**Figure 16 materials-14-05666-f016:**
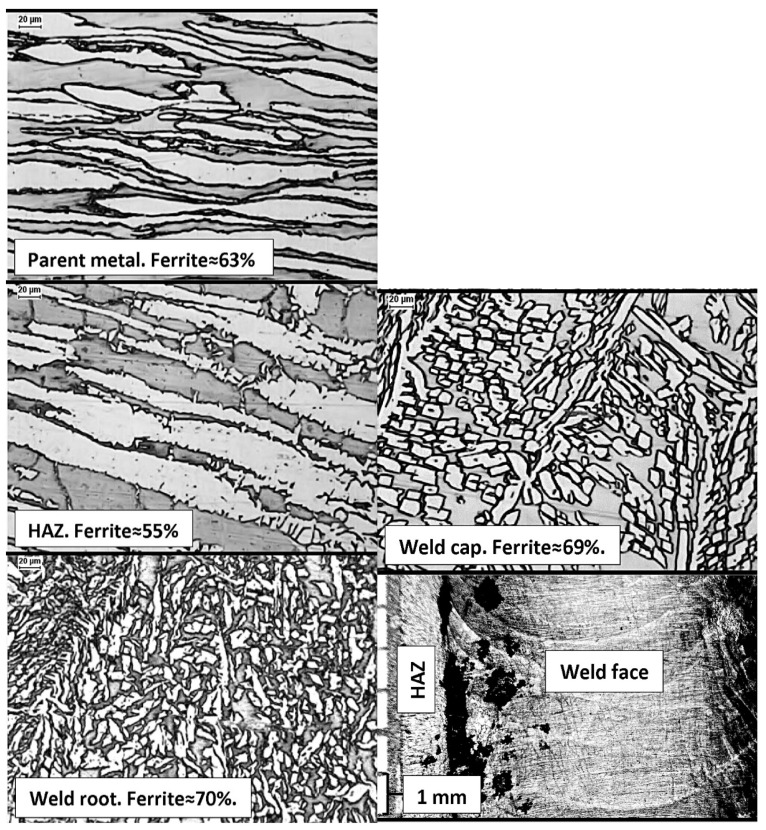
Authors’ analysis of welded connection made of super duplex SDSS steel with a low resistance to pitting corrosion identified for the environment of chlorides.

**Table 1 materials-14-05666-t001:** Maximum solubility of alloy additives in ferrite and austenite (according to [26]).

Alloy Additive	Solubility (%wt.)		Crystal Lattice
	Ferrite	Austenite	
W	35	4.7	A2
Mo	31	1.7	A2
Mn	3.5	100	A1
Cr	100	12.5	A2
Cu	2.1	12	A1
Ni	6	100	A1
Si	11	1.7	A4
C	0.03	2.1	-
N	0.1	2.8	-

## Data Availability

Not applicable.

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
