# Peer review of "Duplex Steels Used in Building Structures and Their Resistance to Chloride Corrosion"

_materials, 2021, doi:10.3390/ma14195666_

Round 1

Reviewer 1 Report

Better to present in abstract briefly some quantitative data

The actual conclusion are rather recommendation than proper details of the results achieved – please reformulate them. Other wise now do not show a correct results between results and conclusions   

Most of the references are out of date, please provide more novel one in order to endorse the novelty of this work

The scale bar in Figure 1 is almost invisible.

I suggest to move the Figure from introduction

Your introduction looks more a report part rather then an introduction for research paper, therefore it should carefully revised to show clearly the context, what other researcher have made and your contribution to research in order to improve the knowledge

You discuss about Figure 2 in section 2 but it is presented in section 1

The scale bar in Figures 15, 16 is almost invisible.

Apart of two Figure, 15 and 16, there are only literature data. Even so no sure how the author obtained these Figures – are they from other study and incorporated in this work ?  

This paper looks rather a review paper because the author contribution to research strategy is omitted

Also if the authors propose this work as a review they have to indicate clearly it and laso in introduction how this work is structured

If this work is a research paper they have to clearly indicate the methods which they applied to obtain the welding , the optical assessment the sample preparation and so on.

Author Response

The detailed comments are given in the attached docx file.

Mariusz Maslak

Reviewer 2 Report

This is a very good article that provides lots of useful information, both to scientists dealing with theory of the HSS welding and practitioners executing welding on real structures.

The following remarks are mainly aimed in resolving some ambiguities in understanding the text and removing the grammatical (or even) style errors in English language.

  • Majority of figures are taken from reference [1] (authored by the same authors as this text) and are not of the sufficiently good quality.
  • Font size in fractions (for the ratio given in the text) should be of the same size as for the regular variables in equations – lines 63, 256, 333, 439 and 864.
  • Page 2 – Figure 1 – top part –there is no scale for size of the microstructure constituents.
  • Page 14 – Caption of Figure 12 – lines 389-390 – improper referencing (see next comment).
  • Text "all photos shown here are taken by the authors" is usually not put in the figure captions. That goes without saying, if no other reference is cited – lines 390, 833, 857.
  • Page 15 – lines 402 and 404 –referring to "dashed lines in the bottom of Figure 12". It should be explained which lines are exactly referred to – there are 4 lines. They should be identified (e.g. with numbers 1, 2, 3, 4) and then referred to them accordingly.
  • Page 15 – line 406 – no precise data is given for any particular steel (like chemical composition) and then there is a statement: "the SDSS weld metal was poor in Nitrogen". Where this data came from? Please, explain.
  • Page 18 – Caption of figure 18 – at the end it contains abbreviations of the used welding procedures. This list would rather belong to some "nomenclature" table given at the beginning or the end of the text.
  • The nomenclature table should be given for there are many abbreviations with which the less experienced reader would not be familiar. Just a suggestion!
  • Equations are part of the text and should be followed by the punctuation mark in the same line.

The scanned pages of the text with the proposed suggestions are enclosed.

Author Response

(The authors gave the same response as above.)

Reviewer 3 Report

The authors of the paper: Duplex steels used in building structures and their resistance to chloride corrosion present a combination of proper data with literature a type of review with proper experimental results also. 

Some corrections must be done in order to improve the paper: 

L26: English corrections are required 

L19-L35: 2-3 references are necessary 

L46: Figure 1 need a scale (mandatory), mention in the picture what are the white and dark areas 

L49: Figure 2 : make it color or use different representations for Ni and Cr (are very close one to each other) 

L91: reference for for Okamoto and L95: reference for Heingart 

L150:improve quality of figure 5 

All figures needs quality improvements 

L172: Table 2 alloing? check English language 

L386: Fig 12: mark the ax X and Y for Cl and Cr variations and why do you choose this area for analyze ? 

L683-687: a reference is required 

L832: Fig 15: all images necessity scales except e) - mention a)-e) for figures and give explanation in L832

L856: scales and a)-e) explanation 

Author Response

(The authors gave the same response as above.)

Round 2

Reviewer 1 Report

-

Author Response

Dear Editor,

A new version of our article has been submitted, with the corrected English language. Moreover, a consent of the editor of the journal "Przeglad Spawalnictwa / Welding Technology Review" to re-publish the figures indicated by the reviewers is attached.

The list of publications cited in the text has been expanded.

We hereby express our special thanks to the special issue editor for the careful editing and linguistic proofreading of our article.

Yours sincerely,

Mariusz Maslak

Cracow University of Technology

Cracow, Poland

Reviewer 3 Report

Publish in current form 

Author Response

Dear Reviewer,

A new version of our article has been submitted, with corrected English language. Moreover, a consent of the editor of the journal "Przeglad Spawalnictwa / Welding Technology Review" to re-publish the figures indicated by the reviewers is attached.

The list of publications cited in the text has been expanded.

We hereby express our special thanks to the special issue editor for the careful editing and linguistic proofreading of our article.

Yours sincerely,

Mariusz Maslak

Cracow University of Technology

Cracow, Poland